# Barriers and Facilitators to the Implementation of a Personalized Breast Cancer Screening Program: Views of Spanish Health Professionals

**DOI:** 10.3390/ijerph19031406

**Published:** 2022-01-27

**Authors:** Celmira Laza-Vásquez, María José Hernández-Leal, Misericòrdia Carles-Lavila, Maria José Pérez-Lacasta, Inés Cruz-Esteve, Montserrat Rué

**Affiliations:** 1Health Care Research Group (GRECS), Department of Nursing and Physiotherapy, University of Lleida-IRBLleida, 25198 Lleida, Spain; celmira.laza@udl.cat; 2Research Centre on Economics and Sustainability (ECO-SOS), Department of Economics, Rovira i Virgili University (URV), 43003 Tarragona, Spain; mariajose.hernandez@urv.cat (M.J.H.-L.); misericordia.carles@urv.cat (M.C.-L.); mariajose.perez@urv.cat (M.J.P.-L.); 3Research Group in Statistical and Economic Analysis in Health (GRAEES), 43204 Reus, Spain; 4Primer de Maig Basic Health Area, Catalan Health Institute (ICS), 25003 Lleida, Spain; icruz.lleida.ics@gencat.cat; 5Department of Basic Medical Sciences, University of Lleida-IRBLleida, 25198 Lleida, Spain

**Keywords:** breast cancer, personalized screening, barriers, facilitators, health professionals

## Abstract

This study explored the barriers and facilitators to the implementation of a risk-based breast cancer screening program from the point of view of Spanish health professionals. A cross-sectional study with 220 Spanish health professionals was designed. Data were collected in 2020 via a web-based survey and included the advantages and disadvantages of risk-based screening and barriers and facilitators for the implementation of the program. Descriptive statistics and Likert scale responses analyzed as category-ordered data were obtained. The risk-based screening was considered important or very important to reduce breast cancer mortality and promote a more proactive role for women in breast cancer prevention, to increase coverage for women under 50 years, to promote a breast cancer prevention strategy for women at high risk, and to increase efficiency and effectiveness. Switching to a risk-based program from an age-based program was rated as important or very important by 85% of participants. As barriers for implementation, risk communication, the workload of health professionals, and limited human and financial resources were mentioned. Despite the barriers, there is good acceptance, and it seems feasible, from the perspective of health professionals, to implement a risk-based breast cancer screening program in Spain. However, this poses a number of organizational and resource challenges.

## 1. Introduction

Systematic reviews of clinical trials assessing the effectiveness of mammographic screening for early diagnosis of breast cancer show a reduction in breast cancer mortality of approximately 20% [1,2,3]. Mammographic screening is also associated with harms such as overdiagnosis, false-positive biopsy findings, or false-negative results that may delay the diagnosis due to false reassurance. Moving from the current one-size-fits-all screening policy to personalized risk-based screening approaches can potentially optimize the harm-benefit ratio, by targeting screening to women who will get the greatest benefit while reducing complications among women who would experience little benefit [4,5].

The personalized screening will probably be the standard for early detection in the near future. Healthcare systems face the challenge of transforming current screening programs to improve their quality and efficiency and contribute to their sustainability. In this regard, it is essential to research the implementation of personalized screening and its implications for clinical practice. A review carried out by Rainey et al. indicates that the implementation of this novel approach is far from straightforward, as it will require a more complex framework with a wider range of stakeholders, spanning multiple healthcare delivery systems and clinical settings [6].

Khan et al. [7] conducted a systematic review that included 10 modeling studies and concluded that risk-based breast cancer screening is cost-effective compared to age-based screening. However, the authors warned of potential biases due to the non-integration of cost and utility parameters for all phases of screening and diagnosis and claimed that more evidence is needed on risk calculation, risk thresholds, screening outcomes by risk categories, and cost and utility parameters. Román et al. [5] carried out a systematic review that included nine modeling studies and an observational study, which showed that screening personalization is effective and efficient. The authors indicated the need for additional studies that assess acceptability, feasibility, and the legal and ethical aspects of personalized screening strategies. In this regard, several authors have expressed the need to consider the views and concerns of health professionals on risk-based screening, because of the influence they may have on the acceptability and successful implementation of a future program [8,9].

Several studies describe the views of health professionals on the implementation of risk-based screening. Professionals’ concerns include the need for specific training [9]; the provision of human and economic resources [10]; the need for more evidence on effectiveness and efficiency of risk-based screening [8]; time constraints unless other health professionals can participate [8]; and the need of computerized risk assessment and risk management tools, to facilitate the integration of breast cancer screening into clinical practice [11].

In Spain, population-based screening programs target women aged 50–69 years for biennial mammograms. At the same time, opportunistic screening in women younger than 50 years is widely used. This study is part of the DECIDO project, which aims to assess the acceptability and feasibility of offering personalized breast cancer screening and its integration in the usual clinical practice [12]. We recently conducted a qualitative study with focus groups of health professionals to explore the barriers and facilitators of implementing a risk-based breast cancer screening program in our National Health Service. The results of this qualitative study have been used to generate a questionnaire on the same topic addressed to a wider sample of health professionals. This study explored the barriers and facilitators to implementing a risk-based breast cancer screening program from the point of view of health professionals, in the context of the Spanish National Health Service.

## 2. Materials and Methods

### 2.1. Design and Study Population

We conducted an exploratory cross-sectional study with health professionals whose work was or was not directly related to breast cancer screening. Our research team contacted and presented the study to the board of directors of several Catalan and Spanish health-related societies and scientific groups (public health, family and community medicine, and breast specialists). We asked them to invite the society or group members to participate in the study which consisted of responding to a questionnaire with an estimated completion time of 20 min. The study information and the link to the questionnaire were posted on the scientific societies’ web pages or in newsletters. For data protection reasons, we did not have information on the number of potential participants, or their demographic or job characteristics Therefore, a self-selection sampling method where individuals choose to take part in research on their own accord, was used.

Data were collected between July and November of 2020, using a web-based survey. The questionnaire was built on the Typeform platform, https://www.typeform.com/ (accessed on 23 January 2022), in the Spanish and Catalan languages. A pilot test with a convenience sample of 20 participants was conducted and some changes were made based on their suggestions.

A survey sample size of 210 professionals was chosen as appropriate so that 95% confidence intervals of the true proportion responding positively would be approximately 7% on either side of the observed proportion. We closed the data collection when 220 health professionals had completed the survey.

### 2.2. Survey Instrument

Questions were iteratively developed jointly by all authors, based on findings from the literature [8,9,10] and discussion groups on views from 29 health professionals. The questionnaire included an introduction describing the study objective, a thank you for participating, a consent authorization to the use of the information, a general definition of the key concepts of *risk-based screening* and *shared decision-making*, and the following sections:Sociodemographic data: age, gender, professional field (nurse, doctor, other), medical specialty or professional profile, years of practice, type of work center (public, private, both, university, other), type of relationship or employment contract, and work relation with early detection of breast cancer (yes/no);Advantages of risk-based screening for the health of women with an individual risk of breast cancer higher (6 items)/lower (6 items) than the population average;Disadvantages of risk-based screening for women’s health (6 items);Advantages of risk-based screening, in relation to current screening, for the Spanish National Health System (4 items);Barriers (15 items) and facilitators (6 items) for the implementation of risk-based screening;Implementation of shared decision-making in breast cancer screening (12 items);Aspects of the organizational structure to consider for the implementation of a risk-based screening program (9 items);Communication of the benefits and harms of breast cancer screening (7 items);Coordination of the risk-based screening program (3 items);

Except for sociodemographic data, all items were scored on 5-point Likert scales. For the first six sections of the questionnaire, related to the risk-based program, the importance given to the statements was assessed as: 1-unimportant, 2-slightly important, 3-moderately important, 4-important, and 5-very important. For the last two sections -communication of the benefits and harms and coordination of the risk-based screening program-, the level of agreement given to the statements was assessed as: 1-strongly disagree, 2-disagree, 3-undecided, 4-agree, 5-strongly agree.

In addition, the survey included these two questions:Considering the advantages and disadvantages, how important is it for you to move from the current Screening Program to a personalized Breast Cancer Screening Program? Answer: 1 to 5 Likert scale, where 1-very little or nothing and 5-a lot;Given the current Breast Cancer Screening Program, do you think Primary Care should be the gateway to a future personalized breast cancer screening program? Answer: Yes/No.

### 2.3. Data Analysis

Descriptive statistics were used to analyze the participants’ characteristics. Likert scale responses were analyzed as ordered-categorical data with frequencies and proportions represented as stacked bar charts. The likert function of the HH package [13], in the R language [14] was used. The exact binomial test was used to obtain a 95% confidence interval for the proportion of participants that rated as “important” or “very important” moving from the current screening program to a personalized program.

We did not assess the validity and reliability of the questionnaire because we did not intend to measure any multifaceted construct, or to obtain a combined score or scale.

## 3. Results

### 3.1. Participants Characteristics

Of the 220 health professionals that participated in the study, 3 out of 4 were women. The median of age was 53 years, and they had a median of 25 years of work experience. Fifty-two percent of participants were physicians, 37% nurses and 11% had other work profiles. The most frequent participants’ specialties were oncology (1 out of 5), epidemiology or public health (1 out of 5), and family and community medicine (1 out of 7). Nine out of 10 worked in the public sector and 4 out of 10 reported that their work was related to early detection of breast cancer (Table 1).

### 3.2. Advantages and Disadvantages of Risk-Based Breast Cancer Screening

Figure 1 shows the advantages of risk-based screening for women’s health, for women at high or low risk of breast cancer, respectively. For women at high risk, 9 out of 10 participants considered that risk-based screening was an important or very important strategy for reducing mortality from breast cancer, increasing survival of diagnosed women, and promoting a more proactive role of women in breast cancer prevention. Also, 8 out of 10 participants, valued as important or very important an increase in coverage for women under 50 years, a greater number of cancers detected in young women, and empowering women in decision-making.

For women at low risk of breast cancer, 7 out of 10 participants considered that risk-based screening was important or very important for empowering women in decision-making, reducing radiation exposure, and reducing the number of mammograms during life. Around 3 out of 5 participants rated as important or very important reducing the number of over-diagnosed cases, false-positive results, and the anxiety associated with waiting for mammogram results.

Figure 2 presents the disadvantages of risk-based screening for women’s health. Two out of 3 participants valued as important or very important the anxiety generated by knowing their risk of breast cancer, for women at high risk. Around 1 out of 2 women think that lack of updating of the ethical criteria and lack of regulation of the legal aspects of the identification of the individual risk is important or very important. One out of 3 gives importance to the resistance that risk-based screening will generate among low-risk women, and to the stigmatization that high-risk women will have. Nevertheless, only 1 out of 6 consider that risk-based screening will socially discriminate against high-risk women.

Figure 3 (top) shows the advantages of risk-based screening for the Spanish National Health System. Nine out of 10 participants valued it as important or very important that risk-based screening may promote a breast cancer prevention strategy for high-risk women. A similar frequency of participants considers that risk-based screening will increase efficiency and effectiveness. Three out of 4 give importance to the increase in coverage for women under 50 years. Figure 3 (bottom) shows that for 85% of the participants moving from the current screening program to a risk-based program was important or very important, with a 95% confidence interval (80%, 90%).

### 3.3. Facilitators and Barriers for the Implementation of a Risk-Based Screening Program

As facilitators for the implementation of a risk-based screening program, importance was given to women’s confidence in health professionals, acceptance of personalized screening by health professionals, positive perception by health professionals of carrying out an activity that benefits women, with frequencies of 9 out of 10 participants (Figure 4). The growing autonomy of women in making decisions that affect their health, acceptance of screening programs by the population, and the experience of health professionals in current screening programs were mentioned as important or very important for more than 7 out of 10 participants.

Regarding barriers to the implementation of a risk-based screening program (Figure 5), 7 out of 10 participants considered important or very important the workload of health professionals; the limited human and financial resources of the National Health System; and the lack of coordination between public and private health care professionals. The barriers rated as least important were the lack of scientific evidence on the effectiveness of risk-based screening; and the resistance of health professionals to the change of the screening model, with only 4 out of 10 participants rating these barriers as important or very important.

Three out of 5 participants valued as important or very important barriers the incorporation of new roles to be developed by health professionals, and the lack of the following: updates to the legal and ethical rules that regulate risk-based screening; adequate computer support; coordination between health professionals at different levels of care; and training of health professionals in risk-based screening.

For 1 out of 2 participants, the following barriers were important or very important: the difficulty of health professionals in communicating with women due to language, cultural and educational barriers; difficulties in including women without public health coverage; little knowledge of health professionals about the benefits and harms of risk-based screening; and difficulty of health professionals to communicate the risk of breast cancer.

### 3.4. Implementation of Shared Decision-Making in Breast Cancer Screening

Figure 6 shows the views of health professionals on the implementation of shared decision-making in breast cancer screening. Nine out of 10 participants considered important or very important the training of health professionals in communication skills to effectively inform about risk and shared decision-making; providing information to health professionals about breast cancer screening; encouraging the participation of women in decisions that affect their health; informing women about the benefits and harms of breast cancer screening before they decide to participate (or not); having a practical guide to implementing shared decision-making, addressed to health professionals; fostering in health professionals the interpersonal skills needed to establish effective clinical relationships with women; and overcoming the resistance from health professionals to changing towards a more participatory decision-making model.

About 8 out of 10 participants rated as important or very important reducing the difficulties of a significant proportion of women to understand the balance of benefits/harms of screening; developing new skills in health professionals to explore women’s experiences and preferences; organizing face-to-face encounters of health professionals and women to decide on participation in the screening program; and overcoming women’s resistance to change towards a more participatory decision-making model.

### 3.5. Aspects of the Organizational Structure to Consider for the Implementation of a Risk-Based Screening Program

Figure 7 presents the opinions of the study participants on aspects of the organizational structure to consider. Nine out of 10 participants considered important or very important the training of health professionals in the Personalized Breast Cancer Screening program; having an integrated and friendly information system; having a system for storing genetic material and mammography data; and training of health professionals on the advantages and disadvantages of moving from the current screening program to personalized screening.

Four out of 5 considered important or very important the incorporation into the National Health System of risk assessment in breast cancer screening; generating a practical guide for implementation; involving other health professionals (pharmacy, laboratories, genetics, others) in the Program; and the coordination between the different levels of care in the National Health System.

### 3.6. Communication of the Benefits and Harms of Breast Cancer Screening and Coordination of the Risk-Based Screening Program

When asked about who should communicate the benefits and harms of breast cancer screening (Figure 8, top), before women decide to participate, the study participants agreed or strongly agreed that the health professionals that should communicate the benefits and harms of breast cancer screening, before women decide to participate, should be: professionals working in hospital breast units (9 out of 10); general practitioners, primary care nurses, and gynecologists (4 out of 5); by mail, with a decision aid or leaflet, or nurses case managers or midwife nurses (7 out of 10). Finally, when answering the question “Who should coordinate a risk-based breast cancer screening program for the initial invitation and follow-up?” (Figure 8, bottom), participants agreed or strongly agreed that the Breast Cancer Screening Program should coordinate the risk-based screening program (9 out of 10) followed by Primary Care (3 out of 5) and nurses case managers (3 out of 5).

## 4. Discussion

### 4.1. Summary of Main Findings

This cross-sectional study assessed the acceptability and feasibility of offering personalized breast cancer screening and integrating it into regular clinical practice in the Spanish National Health System, from the perspective of health professionals. For high-risk women, the best-valued benefit of risk-based breast screening was the reduction in breast cancer mortality and the promotion of a breast cancer prevention strategy. Also, for high-risk women, there was some concern about the anxiety caused by knowing their risk. For low-risk women, there was concern about resistance to fewer mammographic examinations. For the National Health System, participants gave importance to the effectiveness and efficiency of risk-based screening, and 85% considered it important or very important to move from the current screening program to a risk-based program.

Among the facilitators of the implementation of a risk-based screening program, importance was given to the trust of women in health professionals, the acceptance of risk-based screening by health professionals, and the positive perception of health professionals carrying out an activity that benefits women. Among the barriers to implementation, importance was given to the workload of health professionals, the limited human and financial resources of the National Health System, and the lack of coordination between public and private health professionals. Less importance was given to the lack of scientific evidence on the effectiveness of risk-based screening and to the resistance of health professionals to changing the screening model.

Implementing shared decision-making in breast cancer screening would require training health professionals in communication skills and encouraging them to use those skills, informing women about the benefits and harms of risk-based screening and encouraging them to participate in decisions that affect their health, developing a practical guide for shared decision-making implementation, and overcoming the resistance of health professionals and women to switch to a more participatory decision-making model. Regarding aspects of the organizational structure to consider, participants gave importance to the training of health professionals, having good information and storage systems for genetic and imaging data, and good coordination of healthcare resources. There was agreement on the fact that the benefits and harms of breast cancer screening could be communicated by different types of health professionals, including doctors and nurses working at Primary Care centers, and on the role of the current Screening Program as the coordinator of risk-based screening.

### 4.2. Comparison with Previous Studies

The views of our study participants on the benefits of risk-based screening for women are consistent with the literature, especially for those who tend to worry or have their reasons for worrying about having breast cancer, for whom knowing their risk would be reassuring [6,8]. Furthermore, in a study that explored the perceptions of women from three European countries, women concurred that a risk assessment should take place when women turn 40, since lifestyle changes become easier at that age [15]. Similarly, professionals from different European countries consider that personalized screening will generate a proactive role for women by allowing them to take control of some of their risk factors, take measures to reduce the risk of breast cancer [9], and give reassurance to women at low or average risk of developing breast cancer [16]. However, and consistent with our results, Evans et al. [17] identified reluctance to discontinue screening in low-risk women in the United Kingdom, unlike those at high-risk who showed high uptake of additional screening. In addition, some authors draw attention to potential psychological consequences like false reassurance for women at low-risk and fatalistic or obsessive thinking for women at high-risk [6,9].

The professionals in our study were quite concerned about the anxiety that a risk-based program can cause in women. Women’s anxiety, as reflected in our study results, has been linked to a higher risk of breast cancer than the average population and also to worry about cancer [18]. However, the literature also recognizes other elements that increase anxiety and worry: an increase in the frequency of mammographic screening, having an increased risk due to non-modifiable risk factors [9], uneasiness about confronting women with unhealthy lifestyles, and the potential impact of risk disclosure on family members [6]. Anxiety can also be a consequence of suboptimal communication with health professionals or a delay in the delivery of results, so it is necessary for screening services to be fast and efficient and for communication to be clear and effective [19]. Some facilitators proposed to include risk communication by professionals that women trust; proper training and education of professionals so they can provide information on the risks and benefits of screening [20], clarifying the uncertainties that risk communication may generate, and engaging women in shared decision-making [8]; as well as timely delivery of screening results [19].

In our study, the trust relationship between women and primary care doctors was considered a strong facilitator for the implementation of risk-based screening. Willems and Bracke [21] explored the decisions of women in the European Union to participate in cancer screening. General practitioners were found to play a crucial role in making referrals for screening, regardless of the country’s screening strategy. In Spain, the participation initiative in breast cancer screening is driven mainly by the family doctor and not by the screening program, demonstrating the important role of these professionals, mainly among women with less education [21]. Another facilitator of risk-based breast cancer screening could be the awareness of breast cancer screening in society, as it is the most widespread and promoted screening in the European population, with frequent awareness campaigns. That would explain the greater social awareness of this program and the higher participation rate in relation to other screening programs [21]. Thus, both factors can be powerful facilitators of the implementation of risk-based screening.

The time that the current screening model has been running represents a complex barrier for the acceptance of risk-based screening by health professionals. To counteract this, we count on the expected positive results of the clinical trials now underway, the development of guidelines for facilitating implementation [6,22], experience with the current screening program, the perception of professionals that risk-based screening is beneficial for women and raising awareness in society of the advantages of a risk-based approach through the mass media [23]. Other facilitators include training in communication skills, empathy, active listening, and non-verbal language [24]; integrating dynamic risk assessment tools into each individual’s medical records to determine risk, update screening recommendations, and communicating these updates to participants and their healthcare providers [25]; and developing country-specific standardized protocols for the assessment and communication of risk, and provision of screening and prevention recommendations [15].

In the literature, there is no agreement on how risk-based screening for breast cancer should be implemented. Studies with health professionals agree that the complexity of the intervention may require the engagement of other health professionals like nurses or other essential personnel in addition to primary care physicians, who are already overloaded with their current essential work [6,8,15]. A Canadian study suggested creating a centralized program with personnel dedicated to coordinating the different actors and actions [8], a role that, in Spain, could be assumed by the population-based screening program. Some European studies conclude that implementations will vary between countries, depending on how healthcare is funded and arranged [6,15].

Proposals to facilitate the tasks of health professionals include self-collection of information and samples by participating women; test results provided in the mail [10]; and creation of helpful online tools to explain test results [8]. However, attention was drawn to the downside of communicating risk based on inaccurate patient self-reports [26] and to the limitations of women’s understanding of the complex notions of risk estimation and genetics [10]. Therefore, nurses may have a greater role in performing pre-test counseling, risk factor assessment, and supervising data collection [27]. Women agreed that those with above average risk should receive feedback through a consultation with a clinician or trained nurse, either by phone, face-to-face, or through the use of modern technology. Women also suggested group meetings to provide additional information on risk and detection/prevention options, and risk feedback from a medical professional with expert knowledge in the field [15]. To overcome women’s linguistic, cultural, and educational barriers to communication, customized educational and decision tools will be necessary [28]. This would also promote access and equitable provision of risk-based screening to all women [22].

Finally, we would like to point out that some factors related to the implementation of risk-based screening, widely discussed in the literature, did not draw much attention from the participants in our study. On the one hand was the referral and follow-up of high-risk women, which makes it necessary to establish access to genetic counselors and specialists [6] and to protect them from discrimination and potential stigma [29]. On the other hand, were the ethical, legal, and regulatory issues that will be needed across the implementation, from health services planning to the storage of data and results [6,29,30]

### 4.3. Strengths and Limitations

As far as we know, this is the first study that explores barriers and facilitators for the implementation of a personalized screening program, among health professionals in Spain. Our findings can be of interest to health planners and healthcare professionals working in countries with National Health Systems that provide breast cancer screening.

One of the strengths of the study is that the design of the questions was based on the views of health professionals expressed in discussion groups, in a previous study. Other strengths come from the web-based survey design and the recruitment through scientific or professional societies, including lower cost, short-term data collection, easy access to the survey, the confidentiality of responses, and confidence in the study team.

However, in web-based surveys with non-random selection procedures, non-response bias may be a major limitation because the sample is not representative of the study population [31]. Our study’s self-selected participants may have particularly strong feelings or opinions about breast cancer screening, or a specific interest in the study or its findings, that may bias the results. We cannot assess the extent and direction of this bias, related to the incidence of non-response and on how non-respondents differ from respondents on variables of interest. We know that women in the medical work area are overrepresented. Data from the Spanish National Statistics Institute on health professionals show that 51.6% of registered physicians and 84.4% of registered nurses are women. In contrast, the figures for our study are 64.3% and 88.7%, respectively. A stratified statistical analysis would have provided relevant information on health professionals’ views by work area and specialty; however, a much larger sample size would have been necessary to obtain accurate and meaningful results.

We assume that the participants were motivated professionals, with different profiles, from different regions of Spain, who worked at different levels of care and had an interest in breast cancer screening, regardless of their opinions or views on risk-based screening. The novelty of the study findings and its potential contribution to the implementation of risk-based screening help to overcome these limitations and reinforce the utility of the study.

## 5. Conclusions

The study results show that, despite the existing barriers, there is good acceptance of the implementation of a risk-based breast cancer screening program by health professionals, with the benefits for women at high risk the most important consideration. The findings also point out that the implementation of risk-based screening poses several organizational and resource challenges but seems feasible from the professionals’ perspective.

Unlike other contexts, legal and ethical issues, and the lack of evidence on risk-based screening were not important issues for Spanish professionals, but they did highlight the training needs of healthcare professionals as an important issue.

Understanding the barriers and facilitators for the implementation of risk-based screening, from the perspective of women and health planners in Spain, is a relevant topic for future research that can complement the vision of health professionals and facilitate the implementation of personalized screening.

## Figures and Tables

**Figure 1 ijerph-19-01406-f001:**
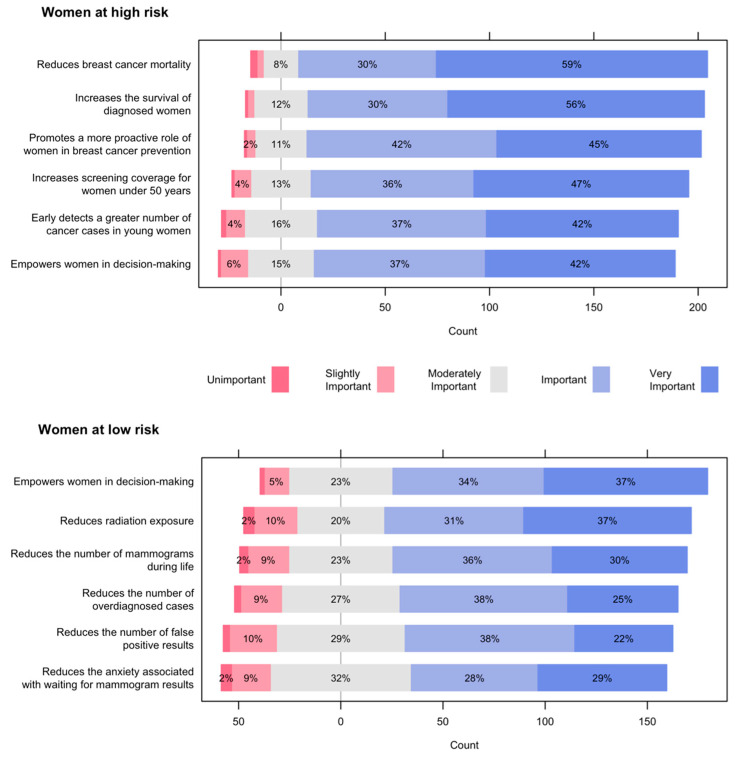
Advantages of risk-based screening for women’s health.

**Figure 2 ijerph-19-01406-f002:**
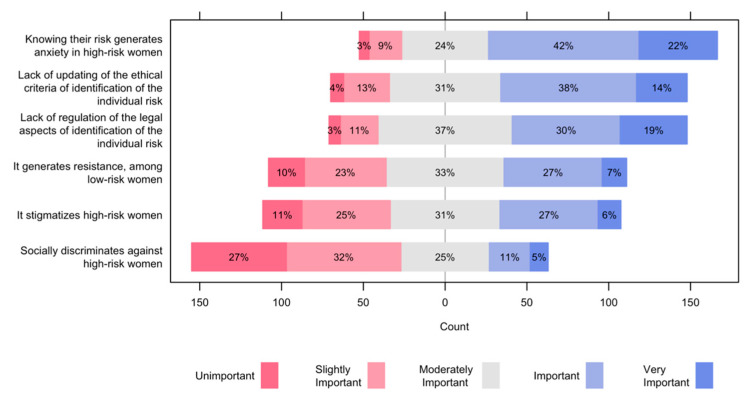
Disadvantages of risk-based screening for women’s health.

**Figure 3 ijerph-19-01406-f003:**
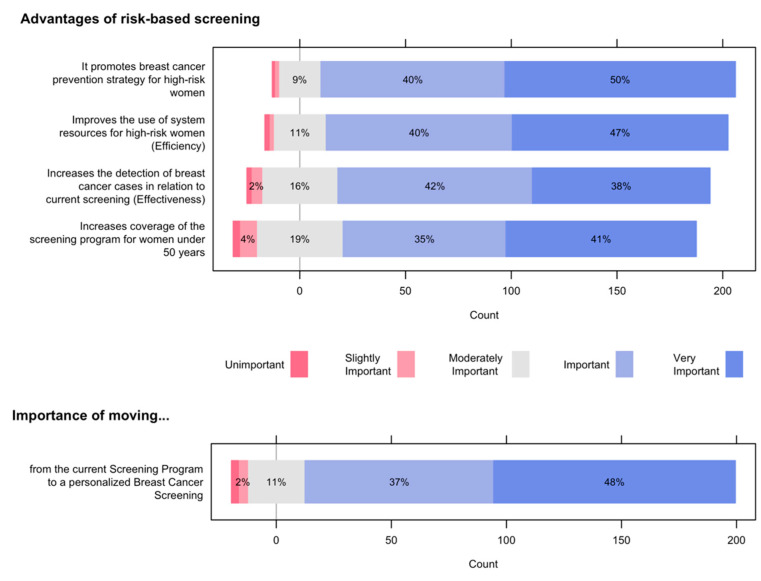
Advantages of risk-based screening for the Spanish National Health System (**top**) and Importance of moving from the current Screening Program to a personalized Breast Cancer Screening (**bottom**).

**Figure 4 ijerph-19-01406-f004:**
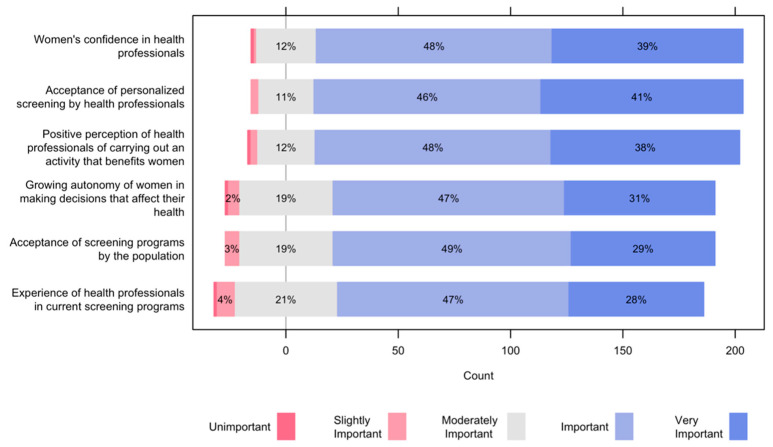
Facilitators for the implementation of a risk-based screening program.

**Figure 5 ijerph-19-01406-f005:**
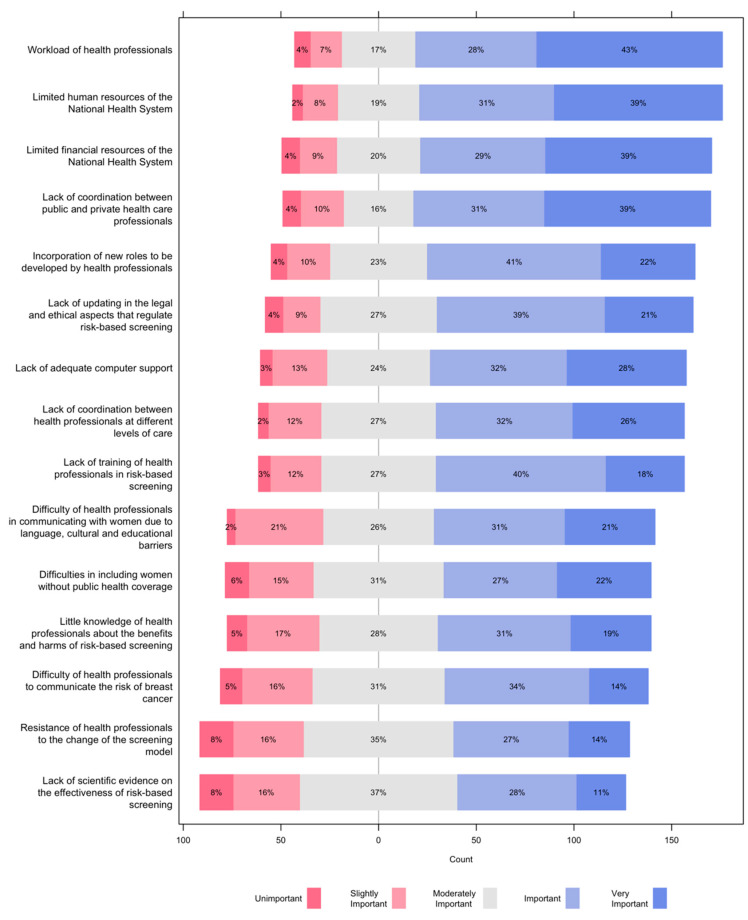
Barriers to the implementation of a risk-based screening program.

**Figure 6 ijerph-19-01406-f006:**
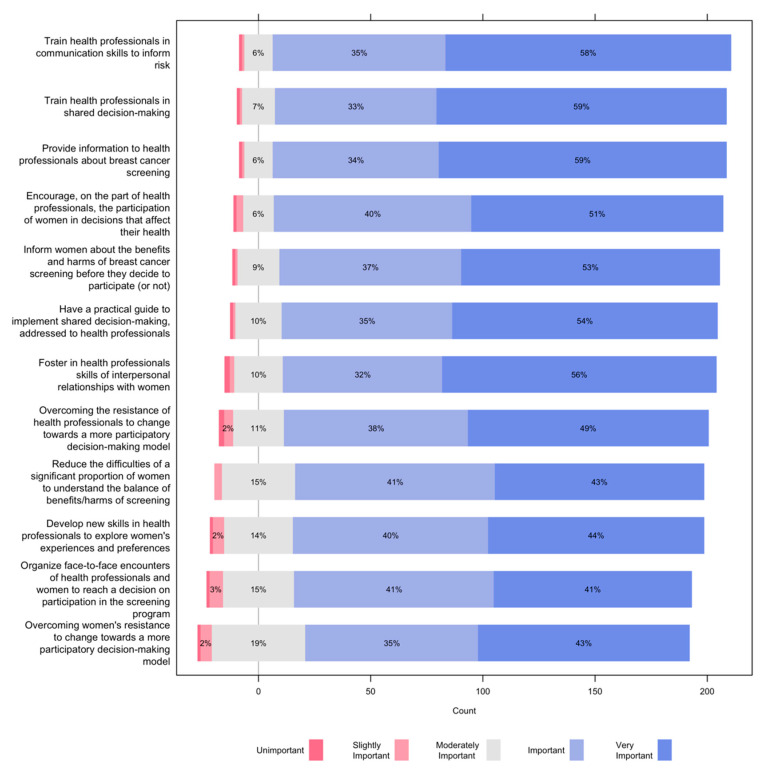
Views of health professionals on the implementation of shared decision-making in breast cancer screening.

**Figure 7 ijerph-19-01406-f007:**
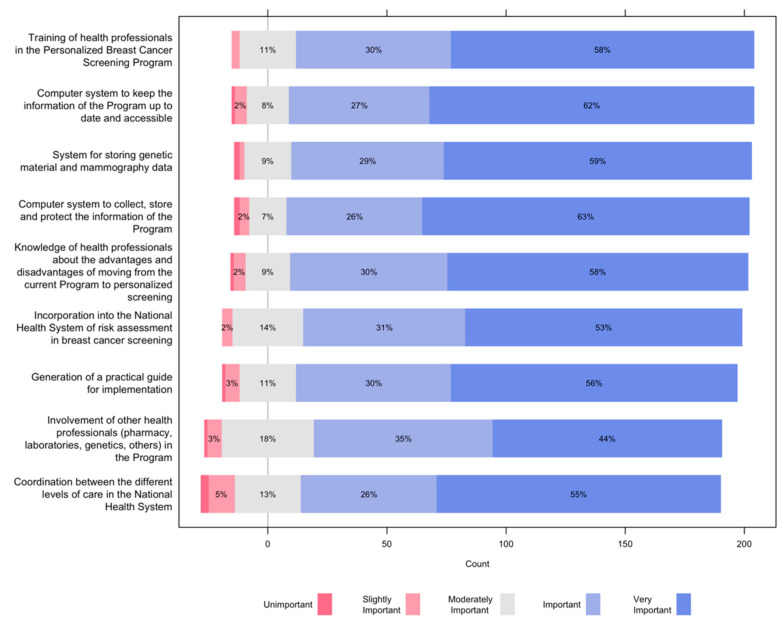
Opinions of the study participants on aspects of the organizational structure to consider for the implementation of risk-based screening.

**Figure 8 ijerph-19-01406-f008:**
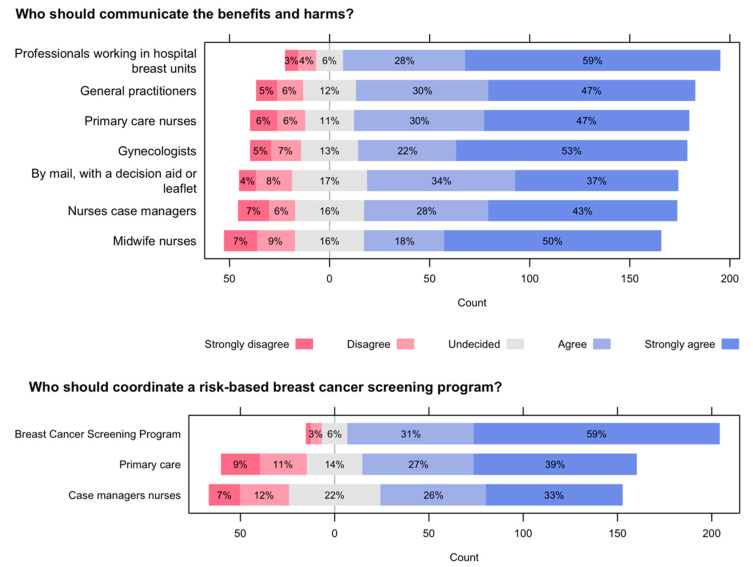
Who should communicate the benefits and harms of breast cancer screening, before women decide to participate? (**top**) and Who should coordinate a risk-based breast cancer screening program for the initial invitation and follow-up? (**bottom**).

**Table 1 ijerph-19-01406-t001:** Demographic and work characteristics of the 220 participant health professionals.

Variable	N (%)	Number of Responses
Gender, Female	151 (76.3)	198
Age, Median [Q1, Q3]	53 [44.8, 60.0]	220
Years of work experience, Median [Q1, Q3]	25.0 [16.0;33.0]	220
Work area		195
Medicine	102 (52.3)	
Nursing	72 (36.9)	
Other	21 (10.8)	
Health-related specialty		195
Oncology	38 (19.5)	
Epidemiology/Preventive Medicine and Public Health	37 (19.0)	
Family and Community Medicine	29 (14.9)	
No specialty	21 (10.8)	
Gynecology and Obstetrics	25 (12.8)	
Radiology	7 (3.6)	
Health economics	6 (3.1)	
Surgery	3 (1.5)	
Other	29 (14.9)	
Workplace		199
Public health center	170 (85.4)	
Private health center	7 (3.52)	
Both public and private health center	10 (5.03)	
University	8 (4.02)	
Other	4 (2.01)	
Work-related to breast cancer early detection	91 (41.6)	219

## Data Availability

The data presented in this study are openly available in the Dryad repository at https://datadryad.org (accessed on 23 January 2022), reference number [doi:10.5061/dryad.hmgqnk9jg].

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
