# Peer review of "Barriers and Facilitators to the Implementation of a Personalized Breast Cancer Screening Program: Views of Spanish Health Professionals"

_ijerph, 2022, doi:10.3390/ijerph19031406_

Round 1

Reviewer 1 Report

I congratulate the authors for this beautiful work. I congratulate him for his previous work in this field. I have a few small suggestions.

Introduction: Well-structured. Research questions should be added.

Method:

How was the population and sample selection made? Country-wide? Is it a city? How many health workers were there? To whom was the form sent? Did everyone get fit? How were 220 people selected? Which probability or nonprobability sampling method was used. It may be important in generalizing the results. It should be explained.

Instead of Variables and information collection, the Ä°NSTRUMENTS title seems more appropriate.

Questions were iteratively developed jointly by all authors, based on findings from 102 the literature (literature should be written)………..how was the validity and reliability of the likert questionnaire created?

The ethical aspect of the study should be added.

Results: It is well-structured, but the spelling of the words at the bottom or top of the figures should be checked.

Discussion: It should be presented in more detail based on the findings. At least three or four paragraphs from the beginning of the discussion should not be presented without reference.

Resources: Current and relevant.

English language editing is required.

Author Response

I congratulate the authors for this beautiful work. I congratulate him for his previous work in this field. I have a few small suggestions.

Answer: Thank you very much.

Introduction: Well-structured. Research questions should be added.

Answer: Our study was exploratory. We did not formulate structured hypotheses, but preferred to state the purpose or research objective. Our research question could have been “What are the views of Spanish health professionals on barriers and facilitators of implementing a risk-based breast cancer screening program in the context of the Spanish National Health Service”? Since including both the research question and the study objective would be redundant, we prefered to maintain the study objective.

Method:

How was the population and sample selection made? Country-wide? Is it a city? How many health workers were there? To whom was the form sent? Did everyone get fit? How were 220 people selected? Which probability or nonprobability sampling method was used. It may be important in generalizing the results. It should be explained.

Answer: The design and study population section has been expanded to address the Reviewer’s questions. Some additional information has also been included in the limitations section of the discussion.

Instead of Variables and information collection, the Ä°NSTRUMENTS title seems more appropriate.

Answer: We agree with the reviewer. We have changed the title to Survey instrument

Questions were iteratively developed jointly by all authors, based on findings from 102 the literature (literature should be written

Answer: References [8-10] were cited.

How was the validity and reliability of the likert questionnaire created?

Answer: We have added the following sentence at the end of section 2.2. Data Analysis: “We did not assess the validity and reliability of the questionnaire because we did not intend to measure any multifaceted construct or to obtain a combined score or scale”.

The ethical aspect of the study should be added.

Answer: In the previous manuscript version we included the Institutional Review Board statement at the end of the manuscript after the Funding statement.

Results: It is well-structured, but the spelling of the words at the bottom or top of the figures should be checked.

Answer: We have checked the words at the bottom or top of the figures. We found an error in the subtitle of Figure 8 (top). Thanks!

Discussion: It should be presented in more detail based on the findings. At least three or four paragraphs from the beginning of the discussion should not be presented without reference.

Answer: We have grouped the discussion paragraphs in subsections (summary of main findings, comparison with previous studies, strengths and limitations). The first three paragraphs refer to the study findings, that’s why we did not include references.

Resources: Current and relevant.

Answer: Thanks!!

English language editing is required.

Answer: The new version of the manuscript has been revised by a native English speaker.

Reviewer 2 Report

The paper analyzes a survey carried out among health professionals concerned with breast cancer, with the intention of detecting the professionals' view of the barriers and facilitators of personalization in the breast cancer screening program in Spain.
Therefore, the study collects the perceptions of the professionals surveyed.
Regarding demography, the high number of oncologists, epidemiologists and nurses interviewed is striking compared to professionals who have direct contact with the target population of population screening: family doctors and gynaecologists, which can bias the results. Likewise, there is an underrepresentation of private centers, which are very relevant at the population level in the large urban centers of the country. It seems to me that it would be interesting to have stratification both by area of ​​work (medicine, nursing, others) and then by specialty. And properly analyze each stratum in a differentiated way.
Given that the professionals who are going to most influence screening techniques are those who initially contact the patients, these assessments should be differentiated.
It should also be noted that much of the interest in the personalization of screening does not come from professionals, but from health authorities as an attempt, sometimes hidden under the new trend of personalizing medicine, to cut back on the cancer screening program breast in Spain, reducing costs for the sake of greater efficiency.
The work would notably improve by stratifying the survey sample by areas of work and by specialty, especially differentiating family doctors and gynecologists from other specialties.
Regarding the methodology, nothing more to add since it is a survey.
The discussion is correct and the bibliography modern and adequately cited.
The conclusions are in line with the intention of the survey, with the reservations already expressed above.

Author Response

The paper analyzes a survey carried out among health professionals concerned with breast cancer, with the intention of detecting the professionals' view of the barriers and facilitators of personalization in the breast cancer screening program in Spain.
Therefore, the study collects the perceptions of the professionals surveyed.

Regarding demography, the high number of oncologists, epidemiologists and nurses interviewed is striking compared to professionals who have direct contact with the target population of population screening: family doctors and gynaecologists, which can bias the results. Likewise, there is an underrepresentation of private centers, which are very relevant at the population level in the large urban centers of the country. It seems to me that it would be interesting to have stratification both by area of work (medicine, nursing, others) and then by specialty. And properly analyze each stratum in a differentiated way. Given that the professionals who are going to most influence screening techniques are those who initially contact the patients, these assessments should be differentiated.

The work would notably improve by stratifying the survey sample by areas of work and by specialty, especially differentiating family doctors and gynecologists from other specialties.
Regarding the methodology, nothing more to add since it is a survey.

Answer: We understand the Reviewer’s concerns. We have expanded the 3rd paragraph of section 4.3. Strengths and limitations of the discussion. We have provided reasons for not stratifying and have discussed the non-representativeness of the sample.

It should also be noted that much of the interest in the personalization of screening does not come from professionals, but from health authorities as an attempt, sometimes hidden under the new trend of personalizing medicine, to cut back on the cancer screening program breast in Spain, reducing costs for the sake of greater efficiency.

Answer: We have not found evidence of the Reviewer’s statement in the literature. All the cited systematic reviews comparing risk-based with aged-based are in favor of risk based, not only for cost-effectiveness but also for a better harm-benefit balance. Moreover, our study participants considered that personalization of screening would need more human and economic resources than the current strategy.

The discussion is correct and the bibliography modern and adequately cited.
The conclusions are in line with the intention of the survey, with the reservations already expressed above.

Answer: Thank you very much.

Reviewer 3 Report

The manuscript reports a well-designed study of expert views on a novel policy with the potential to increase the benefits and reduce the risks of mammographic screening.  It is well-written and relevant to the broad audience of healthcare professionals that reads the International Journal of Environmental Research and Public Health.  I have just a few suggestions:

  • Considering the diversity of the journal’s readership, it may be helpful to add on line 26 of the abstract “from an age-based program” (i.e., “Switching to a risk-based program from an age-based program…”).
  • Section 2.1 of Materials and Methods should include more specific information regarding the distribution of the survey and response rates. For example, was the survey distributed via email, social media, or posted on a website?  If it was distributed via email, were there non-respondents?  If there is no information on non-respondents, is there information on the number of members in the societies and scientific groups that were invited to participate in the study?
  • It is noteworthy that 76.3% of study participants were women. How does that compare with the sex ratios of their healthcare occupations?
  • The Limitations section somewhat addresses the issue of survey non-response. Perhaps it should also note how non-respondents may differ from respondents considering the majority of study participants were female.
  • There are a few spots in the manuscript where wording is a bit confusing. For example, on page 9 line 247, “part” should be changed to “proportion.”  On page 10, the fourth bar from the bottom of Figure 8 reads a bit awkwardly.  On page 10 line 278, “professional profiles” should be changed to “health professionals.”

Overall, the manuscript requires some revision but I recommend it be accepted for publication in the International Journal of Environmental Research and Public Health once the above-mentioned issues have been addressed.

Author Response

The manuscript reports a well-designed study of expert views on a novel policy with the potential to increase the benefits and reduce the risks of mammographic screening.  It is well-written and relevant to the broad audience of healthcare professionals that reads the International Journal of Environmental Research and Public Health.  I have just a few suggestions:

Answer: Thank you very much.

  • Considering the diversity of the journal’s readership, it may be helpful to add on line 26 of the abstract “from an age-based program” (i.e., “Switching to a risk-based program from an age-based program…”).

Answer: Done.

  • Section 2.1 of Materials and Methods should include more specific information regarding the distribution of the survey and response rates. For example, was the survey distributed via email, social media, or posted on a website?  If it was distributed via email, were there non-respondents?  If there is no information on non-respondents, is there information on the number of members in the societies and scientific groups that were invited to participate in the study?

Answer: We have included more details in section 2.1. The survey was announced on web pages and in newsletters. No mails were sent. We did not have a list of members or the number of members that were invited to participate in the study.

  • It is noteworthy that 76.3% of study participants were women. How does that compare with the sex ratios of their healthcare occupations?

Answer: The Reviewer raises a good point. We included some information on the gender distribution of healthcare occupations in section 4.3. Strengths and limitations,

  • The Limitations section somewhat addresses the issue of survey non-response. Perhaps it should also note how non-respondents may differ from respondents considering the majority of study participants were female.

Answer: We have expanded the limitations section, to respond to the Reviewer's comments.

  • There are a few spots in the manuscript where wording is a bit confusing. For example, on page 9 line 247, “part” should be changed to “proportion.”  On page 10, the fourth bar from the bottom of Figure 8 reads a bit awkwardly.  On page 10 line 278, “professional profiles” should be changed to “health professionals.”

Answer: We have changed “part” to “proportion” and “professional profiles” to “health professionals”. We did not figure out which fourth bar from the bottom the Reviewer was mentioning. Figure 8 was on page 11 and had bottom and top subfigures. The fourth bar text of Figure 7 in page 10 did not seem awkward to us.

Overall, the manuscript requires some revision but I recommend it be accepted for publication in the International Journal of Environmental Research and Public Health once the above-mentioned issues have been addressed.

Answer: Thank you very much.